# Costs and cost-effectiveness of community health worker programs focussed on HIV, TB and malaria infectious diseases in low- and middle-income countries (2015–2024): A scoping literature review

James O'Donovan[1,2]*, Cleo Baskin[3], Linnea Stansert Katzen[4,5], Madeleine Ballard[1,2], Maryse Kok[6,7], Ariwame Jimenez[8], Matias Iberico[8,9], Jessica Cook[10], Angele Bienvenue Ishimwe[11], Lily Martin[12], Patrick Kawooya[13], Zeus Aranda[8], Molly Mantus[14], Meghan Bruce Kumar[15,16], Karen E. Finnegan[17,18], Sandra Mudhune[19], Mardieh Dennis[20], Daniel Palazuelos[9,17,21], Dickson Mbewe[22], Michee Nshimayesu[23], Kelsey Vaughan[24]

1 Division of Research, Community Health Impact Coalition, London, United Kingdom, 2 Arnhold Institute for Global Health, Icahn School of Medicine at Mount Sinai, New York, New York, United States of America, 3 Centro de Estudos Estratégicos, Fundação Oswaldo Cruz, Rio de Janeiro, Brazil, 4 Department of Global Health, Faculty of Medicine and Health Sciences, Institute for Life Course Health Research, Stellenbosch University, Cape Town, South Africa, 5 Centre for Health and Sustainability, Department of Women's and Children's Health, Uppsala University, Uppsala, Sweden, 6 Department of Clinical Sciences, Liverpool School of Tropical Medicine, Liverpool, United Kingdom, 7 Malawi-Liverpool Wellcome Trust Clinical Research Programme, Blantyre, Malawi, 8 Compañeros En Salud, Ángel Albino Corzo, México, 9 Tulane University School of Medicine, New Orleans, Louisiana, United States of America, 10 Integrate Health, Boston, Massachusetts, United States of America, 11 TIP Global Health, Kigali, Rwanda, 12 Levy Library, Icahn School of Medicine at Mount Sinai, New York, New York, United States of America, 13 Nama Community Wellness Center, Mukono, Uganda, 14 Last Mile Health, Boston, Massachusetts, United States of America, 15 Department of Nursing, Midwifery and Health, Northumbria University, Newcastle upon Tyne, United Kingdom, 16 KEMRI-Wellcome Trust Programme, Nairobi, Kenya, 17 Department of Global Health and Social Medicine, Harvard Medical School, Boston, Massachusetts, United States of America, 18 Pivot, Ranomafana, Madagascar, 19 Lwala Community Health, Nairobi, Kenya, 20 Last Mile Health, Monrovia, Liberia, 21 Division of Global Health Equity, Brigham and Women's Hospital, Harvard Medical School, Boston, Massachusetts, United States of America, 22 Kasungu District Hospital, Kasungu, Malawi, 23 University of Global Health Equity, Kigali, Rwanda, 24 Bang for Buck Consulting, Amsterdam, Netherlands

* james.odonovan@joinchic.org

## Abstract

Infectious diseases remain a significant public health challenge in low- and middle-income countries (LMICs), with HIV, tuberculosis (TB), and malaria contributing significantly to morbidity and mortality. Community Health Workers (CHWs) play a pivotal role in addressing these diseases, yet evidence on the costs and cost-effectiveness of CHW-led interventions remains fragmented. We performed a scoping review, searching ten databases and the grey literature for original studies published between August 2015 and July 2024. Recognized search terms related to "Community Health Workers" and "Economic Evaluation(s)" in LMICs were utilized. Covidence software was employed to screen studies based on inclusion and exclusion

**Data availability statement:** The data for this study is publicly available at: https://osf.io/h7we8 and in the Supporting Information files.

**Funding:** The authors received no specific funding for this work.

**Competing interests:** The authors have declared that no competing interests exist.

criteria. Data on study methodology, costs and cost-related outcomes were then extracted, tabulated in a data-extraction form, and analysed using Microsoft Excel. **Thirty-three** studies representing 106 scenarios were included, predominantly from sub-Saharan Africa (61%). Over half the scenarios provide evidence about malaria (n = 59), followed by HIV (n = 31) and TB (n = 24). CHWs performed diverse roles, including delivering preventive education, case finding, diagnosis, treatment adherence support, counselling and referrals. The majority demonstrated that CHW programs were cost-effective compared to alternative service delivery models, most commonly facility-based care. These programs were particularly effective in improving treatment adherence and targeting high-priority populations. Costs per beneficiary ranged widely, from $1.20 to $26,556. This review highlights significant heterogeneity in methodologies and reporting, impeding comprehensive comparisons. Future research should emphasize standardized reporting, assess affordability, explore integrated CHW roles across multiple disease groups, and focus on generating evidence that supports priority-setting and resource allocation at the health system level.

## Introduction

Globally, the number of deaths due to infectious diseases s have fallen by half in the last 20 years, from 14.5 million deaths in 2000 to 7.2 million deaths in 2019 [1]. However, substantial health inequalities and inequities remain, with infectious diseases continuing to be major causes of death and illness in low- and middle-income countries (LMICs) [2]. Among the wide range of infectious diseases, HIV, malaria and TB are the three largest contributors to the global disease burden, accounting for over 50 million disability-adjusted life years (DALYs) in 2019 [3]. In 2023, 39.9 million people were living with HIV [4], and 10.8 million people were living with TB [5]. There were an estimated 263 million malaria cases reported worldwide in 2023 [6]. These diseases not only burden populations but also hinder socio-economic development and threaten the achievement of Sustainable Development Goals (SDGs) [7].

The critical role of community health workers (CHWs) in the prevention, control, and treatment of these three major infectious diseases is well-documented, especially in resource-constrained settings [8–11]. CHWs serve as a vital link between communities and formal healthcare systems and provide essential services such as supporting preventive efforts including bednet distribution, indoor residual spraying (IRS) and Integrated Community Case Management (iCCM) for malaria; conducting testing and counselling for HIV and supporting treatment adherence; and referring patients to health facilities as needed for all three diseases [10,12,13].

While CHWs have been around for nearly a century, their emergence as a professional occupational group who are salaried, skilled, supervised and supplied in line with World Health Organization (WHO) guidelines is gaining momentum [14]. CHWs are not merely filling gaps left by nurses and doctors, but are crucial actors in their

own right. By bringing services closer to communities, they help enhance access to care and ensure continuity of service. As such, CHWs have the potential to strengthen health systems, making them more resilient to infectious disease threats and better positioned to achieve long-term health outcomes.

While the literature suggests that CHWs offer great potential for the prevention and management of HIV, malaria and TB in LMICs [15,16], a systematic and comprehensive evaluation of the costs and cost-effectiveness of their work has been lacking. The most recent study to broadly review the costs and consequences of CHW programs in LMICs was a scoping review by Vaughan et al., (2015) [17]. Identifying 36 economic evaluations of various CHW programs, it concluded that CHWs may be a cost-effective approach in some settings. However, this review did not focus specifically on infectious diseases and found only seven malaria, six TB and two HIV related studies. Vaughan et al., (2015) also highlighted significant heterogeneity and methodological challenges in existing economic evaluations, making it difficult to draw definitive conclusions about the cost-effectiveness of these programs.

This current study aims to address this gap in the literature by providing an updated overview of the evidence on the costs, cost-effectiveness and affordability of CHW programs for HIV, malaria and TB in LMICs between 2015–2024. Additionally, it assesses the methodologies used in these evaluations and examines how costs, cost-effectiveness and affordability are reported. By fulfilling these objectives, this research intends to enhance the understanding of the economic value of CHW programs supporting management and prevention of these three diseases, and to support evidence-based decision-making for community health system strengthening.

## Methods

### Nature of review

An initial wider scoping review was conducted to identify and map the available evidence on economic evaluations of both vertical and integrated horizontal CHW programmes in LMICs published between 2015–2024. The protocol was uploaded to Open Science Framework (OSF) on July 27, 2023 [18]. Due to the large number of studies identified and the heterogeneity between studies, reporting of results has been divided into several publications, by disease area or type of CHW, for clarity and to facilitate comparisons between similar studies [19,20]. This paper focuses exclusively on three infectious diseases: HIV, malaria and TB - three of the leading infectious diseases in LMICs [3]. While other infectious diseases, such as COVID-19 and Ebola, have significant public health impacts, they are not included in this study due to the distinct epidemiological contexts, response mechanisms, and funding priorities these diseases demand, and the limited number of studies identified for these other topics.

A scoping review was chosen given the broad and varied nature of the field, with the goal of identifying updated evidence, mapping research methodologies, and highlighting knowledge gaps. This study was conducted in accordance with the Preferred Reporting Items for Systematic Reviews and Meta-Analyses extension for Scoping Reviews (PRISMA-ScR) guidelines [21]. The PRISMA-ScR Checklist is available in the Supplementary Material (S1 Checklist).

### Search strategy and study selection criteria

An initial search was conducted covering January 1, 2015 to July 11, 2023 in the following databases: Ovid MEDLINE(R) and Epub Ahead of Print, In-Process, In-Data-Review & Other Non-Indexed Citations and Daily (1946 to July 06, 2023); Ovid Embase Classic+Embase (1947–2023 July 07); Ovid APA PsycInfo (1806 to July Week 1 2023); Ovid Global Health (1910–2023 Week 26); Ovid AMED (Allied and Complementary Medicine) (1985 to June 2023); Cochrane Central Register of Controlled Trials (CENTRAL); Cumulative Index to Nursing and Allied Health Literature (CINAHL); Web of Science Core Collection; Scopus; and Latin American and Caribbean Health Sciences Literature (LILACS). To ensure this review was up-to-date, a second repeat search was conducted to capture relevant literature up to and including July 16, 2024.

Additionally, we searched the following sources to identify any relevant grey literature: Google Scholar, Bielefeld Academic Search Engine (BASE), DART-Europe E-theses Portal; e-theses online service (EThOS), Open Access Theses and Dissertations, and The OAIster database, plus websites of key organisations involved with CHWs (e.g., CHW Central, Community Health Impact Coalition, and Healthcare Information for All (HIFA.org)). Grey literature included (but was not necessarily limited to) theses or dissertations, preprints or unpublished research, and internal reports.

The search strategy included all appropriate controlled vocabulary and keywords for 'Community Health Workers', 'Economic Evaluations' and 'LMICs', which are defined below. Reference lists of included studies were reviewed to identify any additional studies missed by database searches. Full search strategies are available in the Supplementary Material (S1 Table).

## CHWs

For the purpose of this review we drew upon previous literature [22–24] to define CHWs as healthcare workers who:

(a) are primarily based in the community providing primary healthcare services, both in facilities and in community settings;

(b) are part of the health system (i.e., government or non-governmental organization supported CHWs), performing tasks related to health-care delivery, and/or health education, promotion, or care coordination; and

(c) have received organised training and/or certification, but do not have a tertiary-level degree such as a nursing or midwifery degree.

## Economic evaluations

Both full and partial economic evaluations were included. Full economic evaluations, as defined by Drummond et al., (2015) [25], compare the costs and outcomes of health interventions against alternatives such as the current standard of care or a no-intervention scenario. This may include Cost-Effectiveness Analysis (CEA), Cost-Utility Analysis (CUA), Cost-Benefit Analysis (CBA), Cost-Minimization Analysis (CMA), Cost-Consequence Analysis (CCA), Social Return on Investment (SROI), Multi-Criteria Decision Analysis (MCDA), Budget Impact Analysis (BIA), and Programme Budgeting and Marginal Analysis (PBMA).

Partial economic evaluations, on the other hand, consider costs and/or consequences without necessarily comparing alternatives or linking costs to benefits. They can include outcome descriptions, cost descriptions, cost-outcome descriptions, effectiveness evaluations, or cost analyses.

While full economic evaluations are preferred for guideline and policy development due to their comprehensive nature [26], partial economic evaluations are valuable for initial program development and in contexts where full evaluations are too costly, particularly in LMICs [27].

## LMICs

The World Bank classification of economies was used to categorise LMIC countries as either 'low', 'lower-middle' or 'upper-middle' income based on the costing date for each respective study [28].

## Inclusion and exclusion criteria

Studies were included if they:

• Primarily evaluated CHW programs, excluding those focused exclusively on other healthcare professionals such as doctors, nurses, or midwives.

• Evaluated vertical CHW programs focused on HIV, malaria or TB (e.g., programmes where CHWs were solely focussed on malaria, for example. For HIV, studies were included where HIV related co-infections, such as cryptococcal meningitis, were reported).

- Provided details of an economic evaluation, including either full or partial evaluations.

- Were published between August 2015 and July 2024, as the previous review on this topic covered literature up to July 2015.

- Evaluated interventions or programs located in LMICs as per the World Bank classification in the year the study was costed.

  Studies were excluded if they:

- Were letters to the editor, commentaries, protocols, opinion pieces, policy briefings, or conference abstracts. Although systematic reviews were excluded, their reference lists were searched for potentially eligible studies.

- Assessed the economic impact of digital add-ons to CHW programs (e.g., mobile phone interventions), as the focus of our review was on the economic evaluation of CHW-led interventions themselves, not digital add-ons.

No restrictions were placed on the time frame of the analysis or language of publication. Although the search was conducted in English, full texts were reviewed in any language. Studies were not excluded based on quality due to the high diversity in study types and the interest in exploring the breadth of available evidence. Full eligibility criteria are detailed in the PICO framework in the Supplementary Material (S2 Table).

**Study screening process**

Following a search of the databases and grey literature by a qualified information search specialist, citations were exported to the Covidence platform [29]. Duplicate results were removed using an automated 'de-duplicate' feature within Covidence.

Given the high number of studies identified for screening a team of 18 researchers took part in the initial study screening process. Following training and piloting, each title and abstract was reviewed by two reviewers independently. Any conflicting screening decisions were resolved by a third reviewer who read the study in full and evaluated it against the inclusion and exclusion criteria.

The full texts of the remaining relevant articles were then analysed by two reviewers for final inclusion or exclusion. If a full text was excluded at this stage, a reason was documented. Any conflicts at this stage were flagged within Covidence and resolved by a third reviewer.

**Data extraction**

Data was extracted by two reviewers into a custom Google Sheets document. Each reviewer performed quality control on the other reviewer's extractions, and a third reviewer (an economist) was available to resolve any disagreements. The extraction form was tested for user-friendliness and completeness by all extractors independently and discussed during a joint video conference call. Modifications were made based on the feedback. The spreadsheet captured the article meta-data, information about the study site and CHWs involved in the study, methodological and reporting data, as well as outcomes and cost data.

Outcomes were categorised into five categories: (i) Service Provision (e.g., visits, number of medications distributed, number of household visits); (ii) Population Coverage (e.g., households covered); (iii) Mortality(e.g., reduction in mortality, lives saved) and Morbidity (e.g., TB-related hospitalisations) outcomes; (iv) Cost Savings and Cost Recovery outcomes (e.g., amount of money saved); and (v) Societal Outcomes (e.g., economic growth).

In terms of cost data, the documentation included whether costs were reported in the following categories: (i) Cost per CHW; (ii) Cost per Consultation; (iii) Cost per Service; (iv) Cost per Capita; or (v) Cost per Beneficiary. We also extracted other cost reporting which is specific to infectious diseases, such as Cost per Person/Partner Tested. We documented whether cost per outcome was reported in the study (i.e., Cost per Disability-Adjusted Life Year (DALY) Averted or Cost per Quality-Adjusted Life Year (QALY) Gained). All costs were converted to and reported in 2024 US$ to facilitate

comparison. For costs reported in US$, we first converted costs to local currency units (LCUs) of the same year using that year's exchange rate (World Bank, 'US$ per LCU, period average'). With the costs reported in US$ now in LCUs, and for costs originally reported in LCUs by the resource, we inflated costs to 2024 LCUs using LCU inflation rates reported by the International Monetary Fund ('inflation, average consumer prices'). With all costs in 2024 LCUs, we converted costs to 2024 US$ using the 'LCU per US$, period average' official exchange rate for 2024.

We also report on whether the study authors drew conclusions on the cost-effectiveness and affordability of the CHW program. To determine cost-effectiveness, we looked for comparisons against: (i) thresholds (willingness to pay or gross domestic product (GDP)/capita); or (ii) an alternative service or delivery modality, such as facility-based care. For affordability, we noted whether authors reported how the intervention affects the overall healthcare budget (budget impact analysis), including whether the intervention is affordable within the current budget constraints. That said, we report on cost-effectiveness and affordability based on the authors determination or conclusions from the original study, regardless of whether a threshold was used.

We used Microsoft Excel to organise and analyse extracted data.

### Patient and public involvement

Patients and the public were not consulted as part of this scoping review.

### Ethics approval

A self-assessment was conducted via the University of Washington Human Subjects Institutional Review Board (IRB) which determined that this study was not human subjects research and did not require IRB review.

## Results

### Search results

The initial broader literature search (which included HIV, malaria and TB but also other health areas such as non-communicable diseases (NCDs), mental health, neglected tropical diseases (NTDs) and maternal, newborn and child health (MNCH) as well as horizontal, integrated CHW programs) yielded 9,790 articles, which were reduced to 5,663 after the removal of duplicates. 5,345 studies were excluded following abstract screening, and an additional 170 were excluded after full-text review. After coding studies by disease area, this process resulted in 33 HIV, malaria and TB studies being included in this review, representing 106 scenarios. Further details can be found in the PRISMA flow chart (Fig 1).

Our findings on horizontal integrated CHW programs and NCD and mental health-focused CHWs are currently under review, while we expect to publish the MNCH and NTD findings in 2025 [19,20].

The following three subsections present cost and cost-effectiveness findings separately for malaria, HIV and TB. Each section describes the CHW programs and alternatives assessed and reports on the relevant cost, cost-effectiveness and affordability findings. For cost-effectiveness, we report the incremental cost-effectiveness ratios (ICERs) converted to 2024 US$. An ICER reports the difference in total costs (incremental cost) of the CHW program and the comparator divided by the difference in the chosen measure of health outcome or effect (incremental effect) to provide a ratio of 'extra cost per extra unit of health effect'.

### Malaria

We identified nine studies focused on malaria, with 51 individual scenarios reported within these studies. These studies were conducted in twelve countries (eight low-income and four lower middle-income countries) across two regions. Six of the nine studies were partial cost effectiveness analyses, and three were full economic evaluations (see Table 1).

Malaria interventions were heterogeneous in design, with CHW workforces ranging between 822–47,238 (median: 2,440; using the midpoint of the reported range in the study reporting on multiple countries) and CHW-to-population

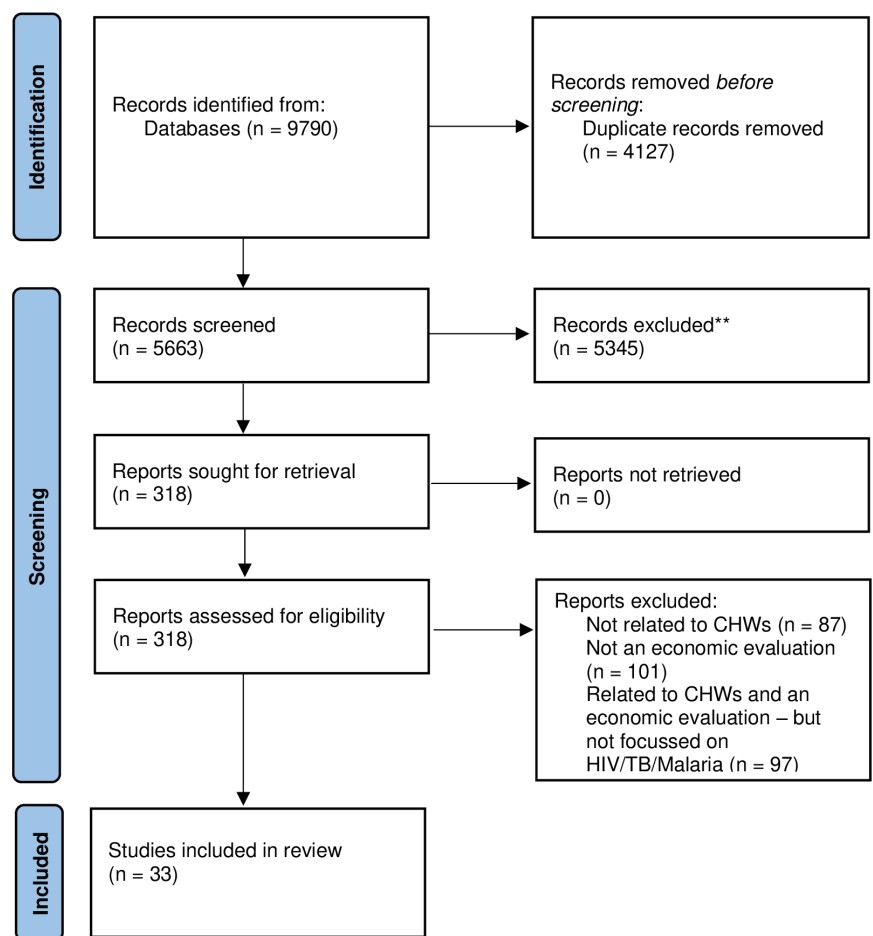

**Fig 1. PRISMA flow diagram.**

ratios spanning from 1:47 beneficiaries to 1:577. CHWs were involved in a wide range of malaria-related service delivery activities, including distributing bed nets (n = 2) [30,31], conducting indoor residual spraying (IRS) (n = 1) [32], distributing prophylactic medication (n = 3) [33–35], and screening, treatment and referrals (n = 3) [30,36,37]. The majority of studies compared the CHW intervention against usual care, more specifically malaria treatment provided by CHWs vs. at health facilities, or in the case of IRS, against standard spraying [32]. Two studies also compared CHW distribution of bed nets to alternative distribution channels [30,31].

Compensation of CHWs varied. In four studies CHWs received stipends [31,33,35,36] (although the specific amounts were only reported in one study) [36]; in one CHWs were salaried with the salary range across different settings being $41.4–165.63 per month/CHW [38]; and one where CHWs received performance based incentives, though the specific amounts were not provided [37]. Finally, the compensation model was not reported in two studies [30–32], and in one, the CHWs were volunteer (see Table 2 for full details) [34].

The most frequently reported cost metric was cost per beneficiary of the intervention, and costs ranged from $1.20 to $104.80 (n = 6 studies). The type of beneficiary varied due to the nature of the intervention. Four studies reported on cost per service [31,33,34,37]. Other cost outcomes reported included cost per structure sprayed [32], cost per net distributed [31], cost per CHW [36], cost per person protected [32], cost per years of life lost due to premature mortality (YLL) averted [30] and ICERs per YLL [30] and per DALY averted [38].

**Table 1. Details of CHW roles and scenarios in malaria intervention studies.**

| Intervention Description | Scenarios descriptions | Role of CHW | Comparator |
|---|---|---|---|
| Effectiveness of seasonal malaria chemoprevention at scale in west and central Africa: an observational study (2020) [35] | | | |
| Seasonal malaria chemoprevention (SMC) in 7 countries | 14 scenarios reporting outcomes across countries (Burkina Faso, Chad, The Gambia, Guinea, Mali, Niger and Nigeria) in 2015 and 2016 | Distribution of SMC to children under 5 | Before/after comparison |
| Cost effectiveness and resource allocation of Plasmodium falciparum malaria control in Myanmar: a modelling analysis of bed nets and community health workers (2015) [30] | | | |
| Diagnosing and treating malaria in areas with varying levels of accessibility in Myanmar | 8 scenarios across different accessibility regions (Easily accessible; accessible; difficult to access; Very difficult to access) and with either CHWs alone or CHWs and intervention | Diagnosis and early treatment of malaria | CHWs only vs. distribution of nets only vs CHW + distribution of nets vs no intervention |
| Indoor Residual Spraying Delivery Models to Prevent Malaria: Comparison of Community- and District-Based Approaches in Ethiopia (2016) [32] | | | |
| Indoor Residual Spraying (IRS) by health extension workers (HEWs) to prevent malaria in Ethiopia | One scenario | CHWs conducting IRS | Standard district implementation |
| Malaria community health workers in Myanmar: a cost analysis (2016) [36] | | | |
| Diagnosing and treating malaria, | 16 scenarios with varying levels of accessibility (easily; medium; difficult; very difficulties) and incentive levels (no incentive; per test & per month (low); per test & per month (higher); monthly stipend) | Diagnosis and early treatment of malaria | Comparing various incentive models vs each other |
| A cost analysis of the diagnosis and treatment of malaria at public health facilities and communities in three districts in Rwanda (2022) [37] | | | |
| Diagnosing and treating malaria at public health facilities and communities in three districts in Rwanda | One scenario | Screening for and treating malaria | Management at health centres and district hospitals |
| Scaling-up the use of sulfadoxine-pyrimethamine for the preventive treatment of malaria in pregnancy: results and lessons on scalability, costs and programme impact from three local government areas in Sokoto State, Nigeria (2016) [34] | | | |
| Sulfadoxine-pyrimethamine (SP) for the preventive treatment of malaria in pregnancy | Six scenarios in three areas in Nigeria and with 1–3 or 1–4 + doses | Distributing SP house-to-house to eligible pregnant women—administered through directly observed treatment (DOTs) | Counterfactual local government areas (LGAs) (facility only strategy) |
| Large-scale delivery of seasonal malaria chemoprevention to children under 10 in Senegal: an economic analysis (2017) [33] | | | |
| Large-scale delivery of seasonal malaria chemoprevention to children under 10 in Senegal | Two scenarios reporting financial or economic costs | Travelling door-to-door to administer the first dose of chemoprevention each month to children aged 3–119 months and to provide amodiaquine (AQ) tablets for the child's caregiver to administer on the subsequent 2 days | N/A |
| Cost effectiveness of pre-referral antimalarial treatment in severe malaria among children in sub-Saharan Africa (2017) [38] [8] | | | |
| Distributing pre-referral antimalarial treatment among children with severe malaria in Kenya | One scenario | Distributing pre-referral malaria treatment to children | Provision of treatment by health facilities |
| Coverage outcomes (effects), costs, cost-effectiveness, and equity of two combinations of long-lasting insecticidal net (LLIN) distribution channels in Kenya: a two-arm study under operational conditions (2020) [31] | | | |
| Two combinations of long-lasting insecticidal net (LLIN) distribution channels in Kenya | Two scenarios from a health system or societal perspective | Distributing LLINs | Alternative distribution channels |

Table 2. Summary details of Malaria-focused interventions.

| Country | Type of Economic Analysis | Population served | CHWs (#) | Compensation method (2024 US$) | Cost/beneficiary* (2024 US$) | Other cost outcomes (2024 US$)*** | DALY ICER (2024 US$) | Cost-effectiveness conclusion** (threshold used) | Affordability conclusion (criteria used) |
|---|---|---|---|---|---|---|---|---|---|
| Effectiveness of seasonal malaria chemoprevention at scale in west and central Africa: an observational study (2020) [35] | | | | | | | | | |
| Burkina Faso | Partial - Cost analysis | 707,317 – 2,056,169 | 19,428 -47,238 (across all countries) | Stipend (not reported) | $3.50 | n/a | n/a | Cost-effective (none) | Affordable (cost saving for the health system compared to current spending) |
| Chad | Partial - Cost analysis | 268,956 – 514,042 | 19,428 -47,238 (across all countries) | Stipend (not reported) | $6.28 | n/a | n/a | Cost-effective (none) | Affordable (cost saving for the health system compared to current spending) |
| The Gambia | Partial - Cost analysis | 88,748 – 90,925 | 19,428 -47,238 (across all countries) | Stipend (not reported) | $10.49 | n/a | n/a | Cost-effective (none) | Affordable (cost saving for the health system compared to current spending) |
| Guinea | Partial - Cost analysis | 253,252 – 438,123 | 19,428 -47,238 (across all countries) | Stipend (not reported) | $4.83 | n/a | n/a | Cost-effective (none) | Affordable (cost saving for the health system compared to current spending) |
| Mali | Partial - Cost analysis | 875,330 – 1,492,137 | 19,428 -47,238 (across all countries) | Stipend (not reported) | $3.72 | n/a | n/a | Cost-effective (none) | Affordable (cost saving for the health system compared to current spending) |
| Niger | Partial - Cost analysis | 596,355 – 1,050,932 | 19,428 -47,238 (across all countries) | Stipend (not reported) | $3.50 | n/a | n/a | Cost-effective (none) | Affordable (cost saving for the health system compared to current spending) |
| Nigeria | Partial - Cost analysis | 860,497 – 1,909,163 | 19,428 -47,238 (across all countries) | Stipend (not reported) | $5.23 | n/a | n/a | Cost-effective (none) | Affordable (cost saving for the health system compared to current spending) |
| Cost effectiveness and resource allocation of Plasmodium falciparum malaria control in Myanmar: a modelling analysis of bed nets and community health workers (2015) [30] | | | | | | | | | |
| Myanmar | Full - Cost Benefit Analysis | Not reported | Not reported | Not reported | $1.46-$5.95 | n/a | n/a | CHW+ distribution of nets in difficult to access areas is cost-effective; other settings are not (comparison with alternatives) | n/a |
| Indoor Residual Spraying Delivery Models to Prevent Malaria: Comparison of Community- and District-Based Approaches in Ethiopia (2016) [32] | | | | | | | | | |
| Ethiopia | Partial - Cost analysis | Not reported | Not reported | Not reported | $1.76 | n/a | n/a | Cost-effective (comparison with alternative) | n/a |

*(Continued)*

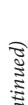

**Table 2.** (Continued)

| Country | Type of Economic Analysis | Population served | CHWs (#) | Compensation method (2024 US$) | Cost/beneficiary* (2024 US$) | Other cost outcomes (2024 US$)*** | DALY ICER (2024 US$) | Cost-effectiveness conclusion** (threshold used) | Affordability conclusion (criteria used) |
|---|---|---|---|---|---|---|---|---|---|
| Malaria community health workers in Myanmar: a cost analysis (2016) [36] | | | | | | | | | |
| Myanmar | Partial - Cost analysis | Not reported | Not reported | Different remuneration options tested: from no incentive, to minimal payment per test or case, to monthly stipend of $38 | Not reported | n/a | n/a | Not reported | n/a |
| A cost analysis of the diagnosis and treatment of malaria at public health facilities and communities in three districts in Rwanda (2022) [37] | | | | | | | | | |
| Rwanda | Partial - Cost analysis | 1,178,077 | Not reported | Other (performance based incentives) | Not reported | Cost per service (episode of standard malaria) $1.54 | n/a | Cost-effective (comparison with alternative) | n/a |
| Scaling-up the use of sulfadoxine-pyrimethamine for the preventive treatment of malaria in pregnancy: results and lessons on scalability, costs and programme impact from three local government areas in Sokoto State, Nigeria (2016) [34] | | | | | | | | | |
| Nigeria | Partial - Cost description | 114,842 | 2,440 | No (volunteer CHWs) | $11.09 | Cost per service (dose) $2.12-$4.46 | n/a | Cost-effective (comparison with alternative) | Affordable (none) |
| Large-scale delivery of seasonal malaria chemoprevention to children under 10 in Senegal: an economic analysis (2017) [33] | | | | | | | | | |
| Senegal | Partial - Cost analysis | 180,000 | 822 | Stipend (not reported) | $1.20 | Cost per service (monthly course administered) $0.46-$0.56 | n/a | Not reported | Likely affordable (compared with curative care costs) |
| Cost effectiveness of pre-referral antimalarial treatment in severe malaria among children in sub-Saharan Africa (2017) [38] | | | | | | | | | |
| Kenya | Full - CEA | Not reported | Not reported | Salaried (range 41.4–165.63 per CHW p/m) | $104.80 | n/a | $7 | Cost-effective (comparison with alternative) | n/a |
| Coverage outcomes (effects), costs, cost-effectiveness, and equity of two combinations of long-lasting insecticidal net (LLIN) distribution channels in Kenya: a two-arm study under operational conditions (2020) [31] | | | | | | | | | |
| Kenya | Full - CEA | 119,246 | Not reported | Stipend (not reported) | Not reported | Cost per service (LLIN distributed) $9.57 | n/a | Not cost-effective (comparison with alternative) | n/a |

*As reported by the authors. Commonly used thresholds such as GDP per capita have faced criticism for failing to consider local resource availability, such as health opportunity costs, and for being less useful in decision-making since it often results in most interventions being labelled as cost-effective.

**Cost per beneficiary defined as the cost per patient treated.

***Only documented here where cost outcome was reported in more than one study for inter-study comparison purposes, or where the cost outcome was used by the authors to determine cost effectiveness.

n/a: not applicable.

As assessed by the study authors, five of the interventions were found to be cost-effective [32,34,35,37,38]. This conclusion was drawn without the use of a threshold or other criteria on one study [35], or on the basis of a comparison with an alternative in four studies [32,34,37,38]. In one study the control arm (with no CHWs) was more cost-effective [31], and in one study one of the scenarios was cost-effective (where CHWs were involved and nets were distributed in difficult to access areas), while scenarios led by CHWs and/or distributing nets in other settings of different remoteness levels were not [30]. Two studies did not assess cost effectiveness [33,36].

Only three studies assessed affordability, all concluding the CHW interventions were affordable [33,34,37]. In two cases this conclusion was reached because the CHW-led intervention was cost saving for the health system compared to current spending [33,35]. In one study authors did not use any criteria to reach this conclusion [34].

## HIV

We identified 14 studies focussed on HIV, with 31 individual scenarios within these 14 studies. The studies were conducted in a mix of upper-middle (South Africa), lower-middle (Kenya and Zambia) and low income (Tanzania, Uganda and Malawi) countries, all of which were in sub-Saharan Africa. The roles of CHWs within these studies included providing administration and logistics support (e.g., mobilizing community health groups), delivering medications, and contacting sexual partners (n = 5) [39–43]; screening for HIV or co-infections (e.g., cryptococcal meningitis) (n = 7) [39,40,43–46] providing ongoing management (e.g., conducting home visits for adherence support) (n = 6) [42,43,45,47–49]; partner notification and tracing (n = 2) [39,42] and providing referrals (n = 9) [39–41,43–47,50]. All studies involved CHWs offering outreach, education, or training to patients, except for one where CHWs' roles were not reported [51]. Twelve studies used standard care as a comparator, typically delivered within a facility (n = 8) (see Table 3).

Table 4 provides further details on included HIV-related studies (see Table 4). The size of the programs in the included studies varied from 6 to 576 CHWs (median: 34) and CHW-to-population ratios spanning from 1 CHW per 12 beneficiaries to 1:41,667 beneficiaries. In four studies, CHWs were salaried, with two reporting the specific monthly salary ($239-$586) [42,45,48,50]. In three studies, CHWs received a stipend, ranging from $10 per patient to a one-off payment of $27 [40,41,49]. Seven studies did not report any details about CHW compensation [39,43,44,46,47,51,52].

Methodologies varied, with nine studies conducting full economic evaluations and five utilizing partial economic evaluations. Cost outcomes demonstrated considerable variation, with the cost per beneficiary (n = 12) ranging from $3.9 to $341 and the cost per new diagnosis (n = 4) ranging from $37.49 to $445.33. Studies conducted in Uganda (n = 2) reported the lowest cost outcomes, with a cost per beneficiary of $3.9 and an ICER per DALY averted of $16.95 [40,49]. Other cost outcomes reported included ICER per patient loss averted (n = 1) [47] and cost per LYG (n = 1) [48].

Eight studies concluded that the CHW-led intervention was cost-effective, with four specifically reporting cost-effectiveness when comparing CHW-led community- or home-based care with standard care delivered at a facility [47,48,51,52]. A study conducted in Tanzania found HIV testing and counselling delivered by CHWs to be most cost-effective delivered in a facility vs. at home or within a public venue [44]. This was due to its integration into existing healthcare services, allowing it to leverage infrastructure, reduce setup and travel costs, and test high volumes of patients efficiently. They found combining facility- and community-based approaches was necessary to achieve widespread diagnostic coverage. Both studies evaluating partner notification services (compared to standard care, delivery by nurses, or passive notification) were considered cost-effective by authors [39,42]. Cost-effective studies often targeted specific high-priority populations (e.g., pregnant women, sexual partners of HIV-positive individuals, adolescents).

Four studies found CHW-led interventions to be not cost-effective [40,41,43,49]. These included home-based HIV testing and counseling and group psychotherapy in Uganda (compared against group HIV education classes), as well as HIV self-testing and ART distribution in Malawi. One intervention in South Africa, integrating NCD screening into home-based HIV testing, yielded inconclusive results [46]. Additionally, one study did not assess cost-effectiveness [50].

**Table 3.** Details of CHW roles and scenarios in HIV intervention studies.

| Intervention Description | Scenarios description | Role of CHW | Comparator(s) |
|---|---|---|---|
| HIV partner services in Kenya: a cost and budget impact analysis study [39] | | | |
| Assisted partner services (aPS): gathering contact information, notifying and locating sexual partners of HIV+ patients in Kisumu, Kenya | Two scenarios reporting outcomes based on whether a high proportion of partners tested positive compared to a lower proportion | Contacting partners about potential exposure and referring to HIV testing and care | 1) Standard care (HIV testing and partner referral alone) 2) aPS model using nurses |
| Expanding HIV testing and linkage to care in southwestern Uganda with community health extension workers [40] | | | |
| Community-based HIV counselling and testing and facilitated linkage to care in Uganda | One scenario | Home-based HIV counseling testing, referral and follow-up | Standard care at public sector facility |
| Methods, outcomes, and costs of a 2.5 year comprehensive facility-and community-based HIV testing intervention in Bukoba Municipal Council, Tanzania, 2014–2017 [44] | | | |
| Facility-, home- and venue-based HIV testing and counseling in Tanzania | Nine scenarios reporting across location (facility; community venue; home) and year (2014; 2015; 2016) | Facility-based: Screening and referral Home- and Venue- Based: counseling services and facilitating the process | • Facility-based • Home-based • Venue-based |
| The effectiveness and cost-effectiveness of community-based support for adolescents receiving antiretroviral treatment: An operational research study in South Africa [47] | | | |
| Community-based support program for adolescents and youth living with HIV who are on ART in South Africa | Two scenarios reporting across year 1 and year 2 of the intervention | Counseling, adherence support, psychosocial interventions, and linkage to necessary services | Standard care at facility |
| Pragmatic economic evaluation of community-led delivery of HIV self-testing in Malawi [41] | | | |
| CHWs performed HIV testing using finger-prick rapid diagnostic tests in Malawi | Two scenarios reporting on HIV self-testing and standard care or standard care alone | Distributing self-test kits, supporting their use, and linking individuals to care and prevention services | Standard care at facility |
| Cryptococcal Meningitis Screening and Community-based Early Adherence Support in People With Advanced Human Immunodeficiency Virus Infection Starting Antiretroviral Therapy in Tanzania and Zambia: A Cost-effectiveness Analysis [48] | | | |
| CHWs supported screening and adherence to cryptococcal antigen treatment in Tanzania and Zambia | Two scenarios reporting different thresholds for CD4 cell count (100 cells/μL; 200 cells/μL) | Weekly home visits to provide adherence support | Standard care at facility |
| The contributions of lay workers in providing home-based treatment adherence support to patients with AIDS in urban settings: lessons from the field in Tanzania and Zambia [50] | | | |
| Screening and home visits for cryptococcal infection in Tanzania and Zambia | Two scenarios in Tanzania or Zambia | Home-based ART adherence support by delivering medications, offering counseling, monitoring treatment side effects, and referrals | Standard care at facility |
| Modeling the cost-effectiveness of home based HIV testing and education (HOPE) for pregnant women and their male partners in Nyanza Province, Kenya [52] | | | |
| Home based HIV testing and education (HOPE) for pregnant women and their male partners in Kenya | Three scenarios reporting differences in HIV status of partner (concordant HIV-; concordant HIV+; discordant) | Conducting home visits for partner counseling | Standard care at facility |
| Assisted partner notification services are cost-effective for decreasing HIV burden in western Kenya [42] | | | |
| Simulation of CHWs providing assisted partner services for HIV testing in Kenya | Two scenarios reporting on HIV status of partner (HIV+; HIV-) | Contacting, testing, counseling and referrals for sexual partners with risk of potential infection | Passive notification |
| Provider-led community antiretroviral therapy distribution in Malawi: Retrospective cohort study of retention, viral load suppression and costs [43] | | | |
| CHWs in community ART distribution in Malawi | One scenario | Traveling from a central health facility to community-based health posts, where they provided refills, testing, and other HIV-related services to stable HIV patients | Standard care at facility |

*(Continued)*

**Table 3.** (Continued)

| Intervention Description | Scenarios description | Role of CHW | Comparator(s) |
|---|---|---|---|
| Economic evaluation of a cluster randomized, non-inferiority trial of differentiated service delivery models of HIV treatment in Zimbabwe [51] | | | |
| Community-based, multi-month ART delivery for HIV treatment in Zimbabwe | Two scenarios based on refill groups at three months or six months | Not reported | 3-month facility dispensing, where patients visited a healthcare facility every 3 months for clinical consultation and a 3-month supply of ART |
| Effectiveness and cost-effectiveness of group support psychotherapy delivered by trained lay health workers for depression treatment among people with HIV in Uganda: a cluster-randomised trial [49] | | | |
| CHWs delivering group support psychotherapy to people living with HIV in Uganda | One scenario | Administering six sessions of psychotherapy | Group HIV education |
| Task-shifting alcohol interventions for HIV+ persons in Kenya: a cost-benefit analysis [45] | | | |
| Salaried CHWs delivering Cognitive Behavioural Therapy (CBT) to HIV patients to reduce alcohol intake in Kenya | One scenario | Providing CBT to reduce alcohol use | Standard care |
| Cost of Integrating Noncommunicable Disease Screening Into Home-Based HIV Testing and Counseling in South Africa [46] | | | |
| Integrating NCD screening and counseling to a home-based HIV counseling and testing program in KwaZulu-Natal, South Africa | One scenario | Screenings for HIV, diabetes, hypertension, and other NCD risk factors, counseling, and referrals | Standard care (home-based HIV testing and counseling) without NCD screening |

Two studies assessed affordability. One concluded that working with CHWs instead of nurses would be affordable, citing cost savings of approximately $421,224 for HIV partner notification services, making it an approach within the national health system's budget [39]. In contrast, the other study determined that home-based HIV counseling and testing exceeded the available public funds for HIV testing in Uganda [40].

## Tuberculosis (TB)

We identified 10 studies, with 24 individual scenarios focused on TB (see Table 5). These took place in eight countries (seven lower-middle income, one low income). CHWs in these studies delivered multiple aspects of care including home-to-home educational visits, screening via collection of sputum samples, and administrative delivery of results and medications. The most common roles of the CHWs were in health service provision, including monitoring, management and treatment (n = 8) [53–60]; screening and case finding (n = 8) [53–55,58–62] outreach and education activities (n = 7) [53–55,57,58,60,62] and referrals (n = 5) [54,55,58,59,62]. To a lesser extent, CHWs were involved in administrative duties such as supply chain and logistics (n = 3) [53,61,62]. Comparators were most frequently facility based care (n = 4) [56,57,59,62].

Of the 10 studies concerning CHW-led TB interventions, the majority were partial economic evaluations consisting of cost analysis (n = 5) [53,54,56,58,61] and cost description studies (n = 1) [55]. The remaining four studies were full economic evaluations, including cost-effectiveness analyses (n = 2) [59,60], a cost-consequence analysis (n = 1) [57] and a social return on investment study (n = 1) [62].

The CHW workforce in these studies ranged from 14-796 workers (median: 134). The CHW-to-beneficiary ratios for TB-focused interventions varied widely, ranging from 1:168 in Uganda to 1:12,262 in Myanmar. Instead of individuals targeted, one study reported the population served in terms of households (n = 470) [61]. The ten studies included salaried CHWs (n = 2, though the salary amounts were not documented) [55,61]; stipend CHWs (n = 3) [56,57,62], where the monthly stipend ranged from $54-$126/month (median $77); and CHWs who received various performance-based payments (n = 3) [53–55]. In some studies CHWs were provided other non-financial benefits (e.g., covering travel costs and

**Table 4.** Summary details of HIV-focused interventions.

| Country | Type of Economic Analysis | Popu-lation served | CHWs (#) | Compen-sation method (2024 US$) | Cost/ benefi-ciary** (2024 US$) | Other cost out-comes *** | ICER DALY (2024 US$) | Cost-effectiveness conclu-sion** (threshold used) | Affordability conclu-sion (criteria) |
|---|---|---|---|---|---|---|---|---|---|
| HIV partner services in Kenya: a cost and budget impact analysis study [42] | | | | | | | | | |
| Kenya | Full-CEA plus BIA | 18,049 | Not reported | Not reported | $13.7 - $16.5 | Cost/ new diagnosis ($37.49- $39.45) | n/a | Cost-effective (comparison with alternative) | Affordable (cost savings of $421,224 compared to using nurses) |
| Expanding HIV testing and linkage to care in southwestern Uganda with community health extension workers [40] | | | | | | | | | |
| Uganda | Partial -Cost description | 126,000 | 62 | Stipend ($39 p/m) | $3.9 | Cost/ new diagno-sis ($177.29) | n/a | Not cost-effective (comparison with alternative-alternative evaluated outside the study) | Not affordable (exceeds current public funds available for HIV testing) |
| Methods, outcomes, and costs of a 2.5 year comprehensive facility-and community-based HIV testing intervention in Bukoba Municipal Council, Tanza-nia, 2014–2017 [44] | | | | | | | | | |
| Tanzania | Partial -Cost analysis | 150,000 | 44 | Not reported | $5.72 - $10.03 | Cost/ new diag-nosis ($ 155.37- $ 445.33) | n/a | Cost-effective delivered in clinics (comparison with alternative) | n/a |
| The effectiveness and cost-effectiveness of community-based support for adolescents receiving antiretroviral treatment: An operational research study in South Africa [47] | | | | | | | | | |
| South Africa | Full - CEA | 6,706 | 576 | Not reported | $40.39 | ICER/ patient loss averted ($489.55- $633.15) | n/a | Cost-effective (comparison with alternative) | n/a |
| Pragmatic economic evaluation of community-led delivery of HIV self-testing in Malawi [41] | | | | | | | | | |
| Malawi | Full - CEA | 24,316 | 347 | Stipend ($10 per patient) | n/a | ICER/ new diagno-sis: $404.59 | n/a | Not cost-effective (willingness to pay) | n/a |
| Cryptococcal Meningitis Screening and Community-based Early Adherence Support in People With Advanced Human Immunodeficiency Virus Infection Starting Antiretroviral Therapy in Tanzania and Zambia: A Cost-effectiveness Analysis [48] | | | | | | | | | |
| Tanza-nia and Zambia* | Full - CEA | 1,001 | Not reported | Sala-ried (not reported) | $339- $341 | Cost/ LYG $76-$99 | n/a | Cost-effective (willingness to pay) | n/a |
| The contributions of lay workers in providing home-based treatment adherence support to patients with AIDS in urban settings: lessons from the field in Tanzania and Zambia [50] | | | | | | | | | |
| Tanzania | Partial -Cost analysis | 1,999 | 6 | Salaried ($586 p/m) | $65.53 | n/a | n/a | Not assessed | n/a |
| Zambia | Partial -Cost analysis | 1,999 | 6 | Salaried ($441 p/m) | $69.94 | n/a | n/a | Not assessed | n/a |
| Modeling the cost-effectiveness of home based HIV testing and education (HOPE) for pregnant women and their male partners in Nyanza Province, Kenya [52] | | | | | | | | | |
| Kenya | Full - CEA | 601 | Not reported | Not reported | $18.29- $20.60 | n/a | 814.65 | Cost-effective (GDP per capita) | n/a |
| Assisted partner notification services are cost-effective for decreasing HIV burden in western Kenya [42] | | | | | | | | | |
| Kenya | Full-CEA plus BIA | 500,000 | 12 | Sala-ried (not reported) | $35.75 - $43.08 | n/a | 1,103 | Cost-effective (GDP per capita) | n/a |
| Provider-led community antiretroviral therapy distribution in Malawi: Retrospective cohort study of retention, viral load suppression and costs [43] | | | | | | | | | |
| Malawi | Partial - Cost analysis | 700 | Not reported | Not reported | $126.44 | n/a | n/a | Not cost-effective (comparison with alternative) | n/a |
| Economic evaluation of a cluster randomized, non-inferiority trial of differentiated service delivery models of HIV treatment in Zimbabwe [51] | | | | | | | | | |
| Zimba-bwe | Full - Cost Benefit Analysis | 4,800 | Not reported | Not reported | $202.78 - $215.42 | n/a | n/a | Cost-effective (comparison with alternative) | n/a |

*(Continued)*

**Table 4.** (Continued)

| Country | Type of Economic Analysis | Popu-lation served | CHWs (#) | Compen-sation method (2024 US$) | Cost/benefi-ciary** (2024 US$) | Other cost out-comes *** | ICER DALY (2024 US$) | Cost-effectiveness conclu-sion** (threshold used) | Affordability conclu-sion (criteria) |
|---|---|---|---|---|---|---|---|---|---|
| Effectiveness and cost-effectiveness of group support psychotherapy delivered by trained lay health workers for depression treatment among people with HIV in Uganda: a cluster-randomised trial [49] | | | | | | | | | |
| Uganda | Full - CEA | Not reported | 60 | Stipend (one-off payment $27) | n/a | n/a | 16.25 | Not cost-effective (comparison with alternative) | n/a |
| Task-shifting alcohol interventions for HIV+ persons in Kenya: a cost-benefit analysis [45] | | | | | | | | | |
| Kenya | Full - Cost Benefit Analysis | 13,440 | 24 | Salaried ($239 p/m) | $44.60 | n/a | n/a | Cost-effective (comparison with alternative) | n/a |
| Cost of Integrating Noncommunicable Disease Screening Into Home-Based HIV Testing and Counseling in South Africa [46] | | | | | | | | | |
| South Africa | Partial -Cost analysis | 570 | 20 | Not reported | $14.38 | n/a | n/a | No conclusion (comparison with alternative) | n/a |

*The cost data is from Tanzania but the health outcomes are from both Tanzania and Zambia.

*As reported by the authors. Commonly used thresholds such as GDP per capita have faced criticism for failing to consider local resource availability, such as health opportunity costs, and for being less useful in decision-making since it often results in most interventions being labelled as cost-effective.

**Cost per beneficiary defined as the cost per patient treated.

***Only documented here where cost outcome was reported in more than one study for inter-study comparison purposes, or where the cost outcome was used by the authors to determine cost effectiveness.

n/a: not applicable.

providing meals) (n = 2) [53,55], or remuneration or benefits were not mentioned (n = 2; n.b. The total does not sum 10 as some studies included CHWs subject to more than one remuneration method) [59,60].

Six of the 10 studies reported the cost per beneficiary, which ranged from $12.57 -$26,556 (median = $8,620). Two studies reported on cost per TB case diagnosed [53,54] ($2 - $6) and three reported cost per patient completing treatment ($30 - $31,999) [53,55,57]. The other studies included cost outcome measures such as cost per contact investigated by CHW [61], cost per patient screened and submitting a sputum sample [53], cost per positive contact diagnosed [54,61], cost per multidrug resistant patient treated [57], and cost per patient re-treated [55]. Only one study, involving two scenarios in Uganda and Cameroon, reported on the incremental cost-effectiveness ratio (ICER) for DALYs averted ($717 and $1,122 respectively) [59]. In half of the studies (n = 5), CHW interventions were reported by authors to be cost-effective when compared against the alternative. In one study, treatment of drug sensitive TB delivered by influential community members was considered cost-effective when compared against a GDP/capita threshold, and the nine-month MDR treatment regimen was found to be more cost-effective than the 20–24 month regimen [60]. The study which looked at delivery by four different iNGOs found the one forming self-help groups that support TB patients by facilitating treatment adherence, providing education, and conducting community mobilization activities to ensure sustainable TB care to be most cost-effective [55]. No studies assessed affordability (see Table 6).

## Methodological findings across all studies

In this section we summarise selected methods-related findings across all included studies (n = 33) and scenarios (n = 106).

Across studies in all three disease areas, the most commonly reported outcomes were cost per beneficiary (n = 23) and cost per service (n = 9). Both metrics varied widely, both within and across disease areas: cost per beneficiary ranged

**Table 5. Details of CHW roles and scenarios in TB intervention studies.**

| Intervention Description | Scenarios description | Role of CHW | Comparator |
|---|---|---|---|
| Role of community health workers in improving cost efficiency in an active case finding tuberculosis programme: an operational research study from rural Bihar, India [54] | | | |
| Existing CHWs in the public health system for active TB case finding in India | One scenario | Screening and referrals, providing patient support (e.g., accompanying patients for testing), advising patients on starting and continuing treatment and raising community awareness | Standard TB detection practices |
| Different challenges, different approaches and related expenditures of community-based tuberculosis activities by international non-governmental organizations in Myanmar [55] | | | |
| Different models of CHW deployment by four international non-governmental organizations (iNGOs) for TB care in Myanmar | Four scenarios reporting outcomes in different international NGOs a.Newly recruited CHVs delivering TB and HIV care for migrants and mobile populations. b.Existing CHVs for hard-to-reach rural populations c.Newly recruited CHVs with diagnostic facilities in rural populations, including internally displaced persons d.Self-help groups supporting TB care in urban slums. | CHWs were involved in different roles depending on the specific scenario: a: health education, active case finding among migrants, and referring suspected cases to health facilities b: Detecting suspected cases, providing referrals, and performing follow-up monitoring in rural hard-to-reach areas c: Operating mobile teams to conduct TB screening, provide health education, and supporting patients through the treatment process d: Forming self-help groups that support TB patients by facilitating treatment adherence, providing education, and conducting community mobilization activities to ensure sustainable TB care | Comparison of four iNGOs |
| Successful expansion of community-based drug-resistant TB care in rural Eswatini - a retrospective cohort study [56] | | | |
| Community-based care for drug resistant TB treatment in Eswatini | One scenario | Delivering daily DOT and intramuscular injectables at the patient's home | Facility-based care led by nurses |
| Examining the cost of community-based tuberculosis treatment in South Africa [58] | | | |
| TB treatment support (e.g., to improve adherence to medications) by CHWs in South Africa | One scenario | Linking patients to treatment supporters for daily adherence support, side effect reporting and subsequent referral to facilities, household contact tracing, and awareness building activities at the district level | National standard of care (traditional care model) |
| Costs and operation management of community outreach program for tuberculosis in tribal populations in India [53] | | | |
| CHWs involved in TB screening, diagnosis and treatment in India | Six scenarios reporting outcomes for screening, diagnosis and treatment services both with top-down and bottom-up costing | Visiting homes and communal areas of the villages systematically to provide TB education, screen, collect sputum samples, and deliver results and TB medication | N/A |
| A cost analysis of implementing mobile health facilitated tuberculosis contact investigation in a low-income setting [61] | | | |
| mHealth-facilitated, home-based strategy for TB contact investigation in Uganda | Two scenarios including top-down or bottom-up costing | Visiting the homes of TB patients, screening all contacts for TB symptoms, and recording their findings using a customized electronic survey application | Routine contact investigation |
| Economic evaluation of a community health worker model for tuberculosis care in Ho Chi Minh City, Viet Nam: a mixed-methods Social Return on Investment Analysis [62] | | | |
| CHW-supported tuberculosis intervention in Vietnam | One scenario | Screening, adherence counseling and psychosocial support, screening at household level, referral of patients with symptoms, collecting and transporting sputum samples. For destitute families, CHWs sometimes provided self-financed nutrition support and transport to the clinic for follow-up consultation | Standard facility-based care |

*(Continued)*

**Table 5.** (Continued)

| Intervention Description | Scenarios description | Role of CHW | Comparator |
|---|---|---|---|
| Cost-consequence analysis of ambulatory clinic- and home-based multidrug-resistant tuberculosis management models in Eswatini [57] | | | |
| CHWs involved in a home-based model for multi-drug resistant TB (MDR-TB) management in Eswatini | One scenario | Administering DOT and injectable treatments, supervising treatment adherence, receiving training in infection control and treatment support, and providing community-based care to reduce the need for patients to travel to healthcare facilities | Facility-based MDR-TB care |
| Cost-effectiveness of community-based household tuberculosis contact management for children in Cameroon and Uganda: a modelling analysis of a cluster-randomised trial [59] | | | |
| Community-based household contact management approach in Cameroon and Uganda | Two scenarios reporting outcomes in Cameroon or Uganda | Conducting household visits for tuberculosis symptom screening, assisting with treatment initiation (done by a nurse), performing follow-up visits to monitor adherence and referring symptomatic children to health facilities for further investigation | Standard facility-based care |
| Economic burden of tuberculosis among Bangladeshi population and economic evaluation of the current approaches of tuberculosis control in Bangladesh [60] | | | |
| Mix of influential community members and/or CHWs treating drug-sensitive (DS) and MDR-TB cases in Bangladesh | Five scenarios including influential community members, CHWs, or a mix of both treating drug-sensitive TB cases, alongside two regimens (9-month and 20–24 month) | Providing DOT, contact tracing, prevention activities | For DS-TB: influential community members vs CHWs vs mix For MDR: 9 vs 20–24 month treatment regimen |

from $1.20 to over $26,000 (median: $10.49), while cost per service ranged from $0.46 to $435 (median: $3.65). Only one study reported Cost per CHW, with scenarios ranging from $2,025 to $5,638 (median: $3,258). There was no clear relationship between CHW remuneration and cost per CHW per year.

In 13 studies, representing the majority of scenarios (n = 58), a provider perspective was used and 17 studies employed a one year time horizon, which is helpful for comparability. All studies included some recurrent costs. However, there was variability in reporting on training costs (n = 26), non-training capital items (meaning items used over one year, such as equipment) (n = 25) and indirect costs or overheads (n = 20). A limited number of studies reported out of pocket and opportunity costs (n = 7) and costs averted (n = 7).

The majority (n = 29) considered cost-effectiveness, most frequently comparing some cost measure for the CHW-led scenario against an alternative (n = 22), such as facility-based care without CHWs. Only 5 studies considered whether the intervention would be affordable in the setting [33–35,39,40].

Only one study used the Consolidated Health Economic Evaluation Reporting Standards (CHEERS) checklist [63], despite it being the leading guidance for authors to follow to ensure that health economic evaluations are identifiable, interpretable, and useful for decision making.

## Discussion

In this current review, we have summarized evidence from 33 studies published between 2015 and 2024 about CHWs involvement in HIV (n = 14), malaria (n = 9) and TB (n = 10) care where details of an economic evaluation were documented. Compared to the 2015 review [17], the number of studies about the costs, cost-effectiveness and affordability of vertical CHW programs focused on HIV, malaria and TB has increased significantly, particularly for HIV given that no evidence was reported on previously. This increase in studies, may reflect the growing role of CHWs in differentiated service delivery and the broader emphasis on cost-effectiveness analyses in global health research.

**Table 6. Summary details of TB focused interventions.**

| Country | Type of Econ. Analysis | Pop. served | CHWs (#) | Compensation method (2024 US$) | Cost/beneficiary* (2024 US$) | Other cost outcome** (2024 US$) | ICER DALY (2024 US$) | Cost-effectiveness conclusion*** (threshold used) | Affordability conclusion*** |
|---|---|---|---|---|---|---|---|---|---|
| colspan Role of community health workers in improving cost efficiency in an active case finding tuberculosis programme: an operational research study from rural Bihar, India [54] ||||||||||
| India | Partial - Cost analysis | 1,021,483 | Not reported | Other - performance and activity linked remuneration | N/A | Cost per person diagnosed ($2.31) | Not reported | Cost-effective (comparison with alternative not evaluated in the study) | n/a |
| Different challenges, different approaches and related expenditures of community-based tuberculosis activities by international non-governmental organizations in Myanmar [55] ||||||||||
| Myanmar | Partial - Cost description | 726, 500-1,434,500 | 117-796 | Various - Salaried (amount not documented; n=1)/ Other (Travel, meals, accommodation; n=2)/ Performance-based (n=1) | $256 - $1,389 | Cost per patient completing treatment ($285-$1,609) | Not reported | iNGO D most cost-effective (comparison with alternatives) | n/a |
| Successful expansion of community-based drug-resistant TB care in rural Eswatini - a retrospective cohort study [56] ||||||||||
| Eswatini | Partial - Cost analysis | 204,000 | Not reported | Stipend ($54/month) | $17,963 | – | Not reported | Cost-effective (comparison with alternative) | n/a |
| Examining the cost of community-based tuberculosis treatment in South Africa [58] ||||||||||
| South Africa | Partial - Cost Analysis | 25 | Not reported | Not reported | $5,674 | – | Not reported | Cost-effective (comparison with alternative) | n/a |
| Costs and operation management of community outreach program for tuberculosis in tribal populations in India [53] ||||||||||
| India | Partial - Cost Analysis | 76,632 | 23 | Other - Monthly travel and communication allowance, plus performance-based payments | N/A | Cost per person diagnosed ($3-$6) Cost per patient completing treatment ($30—45) | Not reported | Cost-effective (comparison with alternative not evaluated in the study) | n/a |
| A cost analysis of implementing mobile health facilitated tuberculosis contact investigation in a low-income setting [61] ||||||||||
| Uganda | Partial - Cost analysis | 470 households | 14 | Salaried (amount not documented) | $11,080-$12,052 | – | Not reported | Unclear (comparison with alternative not evaluated in the study) | n/a |
| Economic evaluation of a community health worker model for tuberculosis care in Ho Chi Minh City, Viet Nam: a mixed-methods Social Return on Investment Analysis [62] ||||||||||
| Vietnam | Full - SROI | 1,465,819 | 151 | Stipend ($126/month) | N/A | – | Not reported | Unclear (social return on investment) | n/a |
| Cost-consequence analysis of ambulatory clinic- and home-based multidrug-resistant tuberculosis management models in Eswatini [57] ||||||||||
| Eswatini | Full - Cost Consequence | 212,531 | Not reported | Stipend ($77/month) | $26,556 | Cost per patient completing treatment ($31,999) | Not reported | Cost-effective (comparison with alternative) | n/a |
| Cost-effectiveness of community-based household tuberculosis contact management for children in Cameroon and Uganda: a modelling analysis of a cluster-randomised trial [59] ||||||||||
| Cameroon | Full - CEA | 3269 | Not reported | Not reported | N/A | – | $717 | Cost-effective (willingness to pay) | n/a |

(*Continued*)

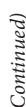

**Table 6.** (Continued)

| Country | Type of Econ. Analysis | Pop. served | CHWs (#) | Compensation method (2024 US$) | Cost/ beneficiary* (2024 US$) | Other cost outcome** (2024 US$) | ICER DALY (2024 US$) | Cost-effectiveness conclusion*** (threshold used) | Afford-ability conclusion |
|---|---|---|---|---|---|---|---|---|---|
| Uganda | Full - CEA | 3311 | Not reported | Not reported | N/A | – | $1,122 | Cost-effective (willingness to pay) | n/a |
| Economic burden of tuberculosis among Bangladeshi population and economic evaluation of the current approaches of tuberculosis control in Bangladesh [60] | | | | | | | | | |
| Bangladesh | Full - CEA | 145-1000 | Not reported | Not reported | $12.57 $2,585 | – | Not reported | Cost-effective: Influential community members (GDP per capita) Cost-effective: 9 month regimen (comparison with alternative (20–24 month regimen)) | n/a |

*Cost per beneficiary defined as the cost per patient treated.

**Only documented here where cost outcome was reported in more than one study for inter-study comparison purposes, or where the cost outcome was used by the authors to determine cost-effectiveness.

***As reported by the authors. Commonly used thresholds such as GDP per capita have faced criticism for failing to consider local resource availability, such as health opportunity costs, and for being less useful in decision-making since it often results in most interventions being labelled as cost-effective.

n/a: not applicable (or data not available).

The large number of studies regarding CHW involvement in vertical infectious disease programs is somewhat surprising, given the WHO international guidance on integrating TB/HIV and malaria/HIV activities dating back to 2004 and 2005 respectively [64,65]. Broader primary healthcare integration started to garner more attention in the 2010s [66,67], though did not reflect prominently in Global Fund programming until the 2023–2028 strategy published in November 2022 [68]. As CHW programs catch up with this guidance, we expect more evidence about horizontal, integrated CHWs in the future. However, this does not diminish the continued relevance of disease-specific studies, which provide targeted evidence to inform programmatic decision-making and funding allocations. Our present review exploring such horizontal programs found 18 such studies published from 2015-2024 [19].

We have reported methods and findings for 106 distinct scenarios, which is perhaps reflective of the range of tasks performed by CHWs, across a variety of geographical settings (ranging from easy- to hard-to-reach) and service delivery locations (community-based locations, homes, and facilities). The majority of evidence included in this review comes from sub-Saharan Africa (n = 25 studies), which aligns with where the burden of disease is concentrated. For example, of the 38.4 million people living with HIV globally in 2021, 67% were in sub-Saharan Africa [69]. Three countries account for half of all scenarios included in this review: Myanmar (n = 36), Kenya (n = 11) and Tanzania (n = 10). In contrast with other literature found as part of the wider review undertaken by our team, including horizontal integrated CHWs who work across multiple disease areas and CHWs working on NCDs [19,20], many of the infectious disease papers reported a large number of scenarios, where for example, different CHW compensation methods, in multiple countries and/or across different geographic areas or service delivery locations, are compared. We hypothesize that this difference reflects the large number of experimental or quasi-experimental studies carried out for infectious diseases. These differences may arise from the substantial funding historically allocated to such vertical programs and the long-established role of CHWs within them. In contrast, task-shifting NCD screening to CHWs is relatively new and subject to distinct epidemiological patterns of disease incidence and prevalence, necessitating tailored approaches to screening and treatment.

Despite the large number of scenarios within each study for each disease (HIV n = 31, malaria n = 59, TB n = 24), there were important differences in study methodologies and the choice of cost outcome. Studies demonstrated a preference for cost outcomes which may have useful programmatic implications, such as cost/person screened, cost/bednet distributed, or cost/CHW/year, but there were not enough comparable outcomes within each disease area, with a small enough range, to be able to draw strong conclusions about the cost of CHW programs in HIV, TB and malaria domains.

Despite this heterogeneity in cost reporting, when comparing CHW-led care to alternative models (e.g., facility-based care or other delivery scenarios without a CHW), most studies found CHW programs to be cost-effective. Twenty-two studies compared some cost measure for the CHW-led scenario against an alternative, and in 25 (53%), the CHW-led scenario was considered more cost-effective than the alternative, with no differences across disease areas. For the remaining, the majority used comparators of CHW-led care in other geographical settings, not care provided by other healthcare professionals. Therefore it is important to highlight that cost-effectiveness can be context dependent, with remoteness level being an important determining factor, with harder-to-reach areas generally incurring higher costs. For HIV, all cost-effective interventions were targeting a specific population group (partners, adolescents, co-infection, pregnant women, patients with alcohol use disorder) as opposed to the interventions screening or providing treatment to the general population, which likely improved the impact of the intervention relative to costs. For malaria and TB, many interventions found to be cost-effective were focused on treatment adherence amongst persons already diagnosed.

From the existing studies identified in this review there is a paucity of data to draw firm conclusions on program affordability. Affordability was not considered in any of theTB-related studies while for malaria just three of the nine studies concluded that CHWs were affordable - although one did not specify the criteria used to reach this conclusion. Two of the 14 studies focused on HIV drew conclusions on CHW program affordability, but had disparate findings. One concluded that CHWs were more affordable than nurses to deliver HIV care [39], while the other determined that CHW delivery of HIV care exceeded currently available public funds [40]. Given the growing evidence that CHW programs are cost-effective,

future research should prioritize budget impact analyses or other methods specifically tailored to assessing affordability, rather than focusing solely on costs and cost-effectiveness. Such tailored approaches would better address the needs of stakeholders, such as government officials, who require actionable insights for resource allocation and program sustainability.

Although it is useful to know that CHW programs for HIV, malaria and TB are generally more cost-effective than alternative delivery modalities using other healthcare professionals, this conclusion may not adequately inform priority-setting at the Ministry of Health level. Officials must navigate complex funding decisions that involve balancing investments between community and facility based initiatives, but also across different disease areas. Comparable outcome measures such as DALYs are needed for this type of comparison, but only featured in five of the studies included in this review. Studies from various LMICs have shown that priority-setting power in health is most strongly held by those with financial control [70]. Given that donors fund the majority of vertical HIV, malaria and TB programs (with the Global Fund alone providing 28%, 76% and 62% of all international financing for HIV, malaria and TB programs, respectively) [71], we suggest that the use of DALYs for priority setting may be at odds with donors' disease-specific priorities and the vertical "silos" they create in global health funding [70,72,73]. The DALY and its use for decision making has been subject to some criticism, though it still features in the latest priority-setting guidance for LMICs [74–80].

Existing CHW investment cases, such as the 2013 One Million CHWs Report [81] and the 2015 CHW Financing Framework [82], have played an important role in shifting the global discourse on CHW programs from a "cost" perspective to an "investment" perspective. However, these reports have been critiqued for lack of methodological rigour and policymakers often question their applicability at the national level. Our review complements these by providing empirical validation for CHW investments beyond broad global ROI estimates and highlights key gaps in affordability data, emphasizing the need for country-specific economic evaluations that integrate real-world financing data, such as national health budgets, donor contributions, and cost-sharing models. This evidence base provides a methodological backbone for policymakers and funders seeking rigorous, empirical validation of CHW cost-effectiveness to inform national decision-making and financing strategies.

## Strengths and limitations

This study has several strengths. Firstly, to our knowledge, it is the only study to specifically document costs, cost-effectiveness and affordability of CHWs working on vertical HIV, malaria and TB programs in LMICs. This review also presents comparable results (in 2024 US$) by disease area, thus facilitating synthesis and comparison of findings. Finally, given the size of the evidence base, we are able to draw some conclusions about the cost-effectiveness of CHWs working in these areas compared to other health professionals, and highlight the importance of geographic setting and context for these results.

The main limitation of this review is that we cannot determine why CHW-led implementation may be more cost-effective than care provided by other health professionals. A common hypothesis is that this cost-effectiveness is driven by differences in remuneration, as CHWs are often compensated at significantly lower levels than other health professionals, if they are paid at all. However, most studies included in this review did not provide sufficient information on the remuneration of both CHWs and other health professionals to confirm this hypothesis. Additionally, we did not extract data on whether the included studies described activities integrated into existing CHW roles or represented standalone activities, though this distinction was often not clearly reported. However, this distinction is important for accurately interpreting costs - such as marginal costs versus total program costs. Finally, we did not assess the quality of the included studies, which constrains our ability to determine which studies offer the most reliable conclusions regarding cost-effectiveness and affordability; however this is in keeping with accepted scoping review methodology [83].

We also only considered the three main infectious diseases (HIV, malaria and TB), while not reporting on other important infectious diseases about which more limited evidence exists. Our results cannot, therefore, be considered representative of all infectious diseases, but only those included in the review.

## Reflexivity statement

We adhered to the consensus statement on equitable authorship in international research collaborations as outlined by Morton et al., (2021) [84]. The following reflexivity statement is provided in that context. This research was conducted by a multidisciplinary, global team comprising researchers and practitioners from LMICs where CHW programs are implemented, such as Malawi, Rwanda, Uganda, Kenya and Liberia. Team members hold positions in academic institutions, non-governmental organisations, and frontline health services (and included CHWs), enabling us to integrate both theoretical and practical insights into our study. All members who contributed to the study design, implementation, analysis, and writing of this paper have been included as co-authors. We acknowledge that the authorship team does not include LMIC government stakeholders, who are a key audience for this research. While many team members have extensive experience engaging with government health systems, this absence may limit study findings. That said, this study was undertaken by the Community Health Impact Coalition (CHIC), a collective of thousands of CHWs and dozens of global health organisations spanning over 60 countries in five WHO regions. The research questions, data collection methods, and analysis were shaped by CHIC's commitment to understanding the drivers of impact and quality in CHW-delivered care globally.

Importantly, this work was also shared with CHWs to explore their opinions and solicit their feedback. Their insights were integral to refining our approach and ensuring the relevance of our findings to those most directly impacted by CHW programs. For a detailed reflexivity checklist, please refer to the Supplementary Material (S2 Checklist).

## Directions for future research

There are several important points that can be addressed by future research. First, stakeholders have expressed a desire for clear conclusions about the cost-effectiveness and affordability of CHWs. Our experience with this review suggests that although definitive conclusions regarding affordability remain limited, evidence on cost-effectiveness is emerging. However, improving the clarity of study reporting is essential to ensure that those who most need this evidence can effectively interpret and apply the findings. In the absence of a CHW-specific reporting checklist, authors are advised to use the CHEERS checklist [63], as adherence to a reporting checklist has been shown to improve reporting quality [85].

Second, and related to reporting, lack of evidence about CHW remuneration, roles and their importance for cost-effectiveness limits our understanding of why CHWs might be more cost-effective, which can inform how to implement CHW programs in the future for maximum cost-effectiveness. Considering recent discussions on the importance of fair compensation for CHWs [86], this information is crucial to include in any discussion around cost-effectiveness. Additionally, future research should examine how workforce shortages and inconsistent integration into national health systems impact the scalability and sustainability of CHW programmes, despite their demonstrated cost-effectiveness.

Third, we suggest future research aligns with WHO care integration recommendations and considers horizontal, integrated CHWs who work on delivering HIV, malaria and TB care, but also across other health areas as well. For this type of study, an outcome measure like the DALY which encompasses both morbidity and mortality improvements across disease areas becomes increasingly important. Researchers need not limit themselves to a single outcome measure and can continue to report on cost per beneficiary and other such measures that can be useful for programmatic planning. However, if we aim to influence priority-setting and resource allocation decisions at the ministerial level and improve comparability of findings across studies, DALYs are currently the best measure to use.

Given that the majority of current evidence is focused on three countries, which is perhaps a reflection of donor research priorities, funding distribution, or the prevalence of health challenges in these countries, it highlights the need for economic evaluations to be conducted in more geographical contexts, to ensure that findings are applicable across a broader range of settings.

Finally, future research should integrate real-world financing data, including donor contributions, national health expenditures, and cost-sharing models, to ensure cost-effectiveness findings are directly actionable for policymakers and funders. Regional collaborations, such as the Africa CDC's ongoing economic review of CHW services across the African

Union, present opportunities to improve CHW financing research [87]. One avenue for translating research into policy is through organizations such as the Community Health Impact Coalition (CHIC), which leverage collaborative initiatives to inform Ministries of Health on CHW model development.

## Conclusion

The evidence base on CHW involvement in HIV, malaria and TB programs has expanded significantly since the Vaughan et al., (2015) review. The current available evidence allows us to conclude that CHWs are often more cost-effective for vertical HIV, malaria and TB interventions than similar care delivered by other health professionals. However, not enough studies assess whether these interventions are affordable to governments. As health service delivery shifts to integrated models, future research is likely to consider more horizontal, integrated CHW programs as opposed to the vertical ones assessed in this review.

## Supporting information

**S1 Checklist.  Preferred Reporting Items for Systematic reviews and Meta-Analyses extension for Scoping Reviews (PRISMA-ScR) checklist.**
(DOCX)

**S2 Checklist.  Reflexivity checklist.**
(DOCX)

**S1 Table.  Search strategies.**
(DOCX)

**S2 Table.  Population, Intervention, Comparison, and Outcome (PICO) framework.**
(DOCX)

## Acknowledgments

We thank the following members of the Community Health Impact Coalition Research team for contributing to, or reviewing the final manuscript prior to publication: Helen Olsen, Rizky Deco Praha, Trio Sirmareza, Josef Ernst, Marius Nkenfack, Adriana Viola Miranda, John Wabwire, Sherlie Petit Homme, Mary Juma, Jane Nelima.

## Author contributions

**Conceptualization:** James O'Donovan, Maryse Kok, Ariwame Jimenez, Jessica Cook, Angele Bienvenue Ishimwe, Patrick Kawooya, Zeus Aranda, Molly Mantus, Meghan Bruce Kumar, Sandra Mudhune, Mardieh Dennis, Daniel Palazuelos, Dickson Mbewe, Michee Nshimayesu, Kelsey Vaughan.

**Data curation:** James O'Donovan, Madeleine Ballard, Lily Martin, Kelsey Vaughan.

**Formal analysis:** James O'Donovan, Cleo Baskin, Linnea Stansert Katzen, Kelsey Vaughan.

**Funding acquisition:** James O'Donovan, Madeleine Ballard.

**Investigation:** James O'Donovan, Cleo Baskin, Linnea Stansert Katzen, Lily Martin, Kelsey Vaughan.

**Methodology:** James O'Donovan, Kelsey Vaughan.

**Project administration:** James O'Donovan, Kelsey Vaughan.

**Resources:** James O'Donovan, Kelsey Vaughan.

**Software:** James O'Donovan, Lily Martin, Kelsey Vaughan.

**Supervision:** James O'Donovan, Linnea Stansert Katzen, Kelsey Vaughan.

**Validation:** James O'Donovan, Cleo Baskin, Linnea Stansert Katzen, Kelsey Vaughan.

**Visualization:** James O'Donovan, Cleo Baskin, Kelsey Vaughan.

**Writing – original draft:** James O'Donovan, Cleo Baskin, Linnea Stansert Katzen, Kelsey Vaughan.

**Writing – review & editing:** James O'Donovan, Cleo Baskin, Linnea Stansert Katzen, Madeleine Ballard, Maryse Kok, Ariwame Jimenez, Matias Iberico, Jessica Cook, Angele Bienvenue Ishimwe, Patrick Kawooya, Zeus Aranda, Molly Mantus, Meghan Bruce Kumar, Karen E. Finnegan, Sandra Mudhune, Mardieh Dennis, Daniel Palazuelos, Dickson Mbewe, Michee Nshimayesu, Kelsey Vaughan.

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
