## [Decision Letter · Decision Letter 0]

14 Feb 2025

PGPH-D-25-00054

Costs and Cost-Effectiveness of Community Health Worker Programs focussed on HIV, TB and Malaria Infectious Diseases in Low- and Middle-Income Countries (2015-2024): A Scoping Literature Review

Dear Dr. O’Donovan,

Thank you for submitting your manuscript to PLOS Global Public Health. After careful consideration, we feel that it has merit but does not fully meet PLOS Global Public Health’s publication criteria as it currently stands. Therefore, we invite you to submit a revised version of the manuscript that addresses the points raised during the review process.

We look forward to receiving your revised manuscript.

Kind regards,

Angela Devine, PhD

Academic Editor

Journal Requirements:

Additional Editor Comments (if provided):

Reviewers' comments:

Reviewer's Responses to Questions

**Comments to the Author**

1. Does this manuscript meet PLOS Global Public Health’s publication criteria ? Is the manuscript technically sound, and do the data support the conclusions? The manuscript must describe methodologically and ethically rigorous research with conclusions that are appropriately drawn based on the data presented.

Reviewer #1: Yes

Reviewer #2: Yes

2. Has the statistical analysis been performed appropriately and rigorously?

Reviewer #1: Yes

Reviewer #2: N/A

3. Have the authors made all data underlying the findings in their manuscript fully available (please refer to the Data Availability Statement at the start of the manuscript PDF file)?

Reviewer #1: Yes

Reviewer #2: Yes

4. Is the manuscript presented in an intelligible fashion and written in standard English?

Reviewer #1: Yes

Reviewer #2: Yes

5. Review Comments to the Author

Reviewer #1: I appreciate the opportunity to review this manuscript. Below, I provide my evaluation based on the significance of the claims, contextualization within the literature, and robustness of the data and analyses. I also offer minor editorial suggestions.

Significance of the claims

The manuscript’s central claim is that CHW programs are generally cost-effective compared to facility-based care, particularly in improving treatment adherence and targeting high-priority populations. It also highlights methodological heterogeneity across studies, which limits direct comparisons. Given the increasing recognition of CHWs as essential to primary healthcare in LMICs, this review is highly relevant for policymakers and donors considering CHW investment.

Placement within the existing literature

The manuscript builds effectively on prior research, particularly Vaughan et al., by incorporating newer evidence and addressing methodological gaps. The literature review is thorough, drawing from both peer-reviewed and grey literature. However, additional contextualization would further strengthen the discussion:

• Comment 1: Engagement with CHW Investment Cases

o The manuscript does not sufficiently engage with existing investment cases for CHWs, many of which include cost-effectiveness analyses. Discussing how these findings align with or differ from such investment cases would add valuable context.

o Several CHW investment cases in LMICs include cost-effectiveness and budget impact analysis results. Have these been considered? While some provide horizontal evidence, others may offer disaggregated results.

o Page 46: The statement—"Given the growing evidence that CHW programs are cost-effective, future research should prioritize budget impact analyses or other methods specifically tailored to assessing affordability, rather than focusing solely on costs and cost-effectiveness"—should acknowledge the substantial number of CHW investment cases already available.

• Comment 2: Recent Africa CDC review

o The Africa CDC Health Economics Division recently conducted a review on the cost-effectiveness of CHW program components. While I have not seen the results, it may be valuable to explore potential synergies between this review and your analysis.

o Expanding on how research gaps (e.g., affordability and integrated CHW models) could be addressed—such as through collaborations with Africa CDC or Ministries of Health—would enhance the paper’s impact.

Do the data and analyses support the claims?

The study follows a transparent methodology for study selection and data extraction. The categorization of cost outcomes is particularly useful for policy interpretation. However, several limitations should be addressed:

• Comment 3: Justification for relying on authors’ cost-effectiveness and affordability conclusions (page 13)

o The manuscript clarifies that it reports only the original authors’ assessments of cost-effectiveness and affordability, noting this as a limitation. However, it would be helpful to briefly justify this decision. Could a reassessment be possible using publicly available data? If not, is this an area for future research? The absence of CHEERS checklists in many studies is mentioned—does this make independent assessment infeasible?

• Comment 4: CHW compensation data

o The manuscript frequently notes that CHW remuneration details are unavailable. However, in some cases, could this information be obtained from publicly available sources such as national health budgets? If this approach was considered and deemed unfeasible, clarifying why would strengthen the discussion.

Minor Edits

• Comment 5: Page 5: The term "disease mortality rates" is used, but the cited data refers to the absolute number of deaths.

• Comment 6: Page 5: Consider adding the term "inequities" alongside "health inequalities", as inequities imply avoidable and unjust differences.

• Comment 7: Page 12: In the section "(iii) Mortality and Morbidity Outcomes," include an example of a morbidity outcome.

• Comment 8: Page 13: Typo: "ee" should be "we."

• Comment 9: Page 28: The manuscript states, "A study conducted in Tanzania found HIV testing and counseling delivered by CHWs to be most cost-effective when delivered in a facility vs. at home or in a public venue." Consider clarifying in the inclusion/exclusion criteria that CHWs can be based in facilities as well as in the community, as this is a common misconception.

• Comment 10: Page 44: The manuscript notes that since the 2015 review, the number of economic studies on vertical CHW programs for HIV, malaria, and TB has significantly increased. It would be valuable to comment on potential reasons—e.g., an increase in CHW-delivered services or a general rise in cost-effectiveness studies?

• Comment 11: Page 45-46: The transition between the last sentence on page 45 and the first sentence on page 46 could be clarified for readability.

• Comment 12: General: There is a significant shortfall of CHWs in LMICs. While there is evidence that CHWs are cost-effective, generate a strong return on investment, and are affordable, they remain underutilized. It may be useful to highlight this in the introduction or discussion.

• Comment 13: Throughout: The paper discusses disputes over cost-effectiveness thresholds. Consider citing: Pichon-Riviere et al. DOI: 10.1016/S2214-109X(23)00162-6 and Ochalek et al. DOI: 10.1136/bmjgh-2018-000964

Reviewer #2: This review is a comprehensive documentation of the available literature on the costs and cost-effectiveness of CHW programs focused on the three infectious diseases in question. The collation of evidence from the studies plus the synthesis narrative allows readers an accessible means of gauging the breadth and depth of evidence available on this topic. The conclusions were supported by the review findings, and the focus on cost-effectiveness and affordability of interventions is of ever increasing importance. I have only a few points for potential correction:

Page 13: there appears to be a typing error at the end of the second paragraph "That said, ee report"

Figure 1: some of the numbers don't seem to add up as I expected, e.g. 5663 records screened minus 5345 records excluded gives 318 rather than the 316, then minus the additional 170 excluded gives a number higher than the final 33 included

6. PLOS authors have the option to publish the peer review history of their article (what does this mean? ). If published, this will include your full peer review and any attached files.

**Do you want your identity to be public for this peer review?** For information about this choice, including consent withdrawal, please see our Privacy Policy .

Reviewer #1: No

Reviewer #2: **Yes: ** Christina Banks

---

## [Decision Letter · Decision Letter 1]

15 Apr 2025

Costs and Cost-Effectiveness of Community Health Worker Programs focussed on HIV, TB and Malaria Infectious Diseases in Low- and Middle-Income Countries (2015-2024): A Scoping Literature Review

PGPH-D-25-00054R1

Dear O’Donovan,

We are pleased to inform you that your manuscript 'Costs and Cost-Effectiveness of Community Health Worker Programs focussed on HIV, TB and Malaria Infectious Diseases in Low- and Middle-Income Countries (2015-2024): A Scoping Literature Review' has been provisionally accepted for publication in PLOS Global Public Health.

Best regards,

Angela Devine, PhD

Academic Editor

Reviewer Comments (if any, and for reference):

Reviewer's Responses to Questions

**Comments to the Author**

1. If the authors have adequately addressed your comments raised in a previous round of review and you feel that this manuscript is now acceptable for publication, you may indicate that here to bypass the “Comments to the Author” section, enter your conflict of interest statement in the “Confidential to Editor” section, and submit your "Accept" recommendation.

Reviewer #1: All comments have been addressed

2. Does this manuscript meet PLOS Global Public Health’s publication criteria ? Is the manuscript technically sound, and do the data support the conclusions? The manuscript must describe methodologically and ethically rigorous research with conclusions that are appropriately drawn based on the data presented.

Reviewer #1: Yes

3. Has the statistical analysis been performed appropriately and rigorously?

Reviewer #1: Yes

4. Have the authors made all data underlying the findings in their manuscript fully available (please refer to the Data Availability Statement at the start of the manuscript PDF file)?

Reviewer #1: Yes

5. Is the manuscript presented in an intelligible fashion and written in standard English?

Reviewer #1: Yes

6. Review Comments to the Author

Reviewer #1: The authors have done a great job addressing all of my comments.

7. PLOS authors have the option to publish the peer review history of their article (what does this mean? ). If published, this will include your full peer review and any attached files.

**Do you want your identity to be public for this peer review?** For information about this choice, including consent withdrawal, please see our Privacy Policy .

Reviewer #1: **Yes: ** Katherine Snyman
